# Assessing changes in mood state in university students following short-term study abroad

Tsukasa Yamanaka[1,2⚬]*, Noriko Yamagishi[2,3,4⚬], Norberto Eiji Nawa[4,5⚬], Stephen J. Anderson[6⚬]

**1** College of Life Sciences, Ritsumeikan University, Kusatsu, Shiga, Japan, **2** Ritsumeikan Inamori Philosophy Research Center, Ibaraki, Osaka, Japan, **3** College of Global Liberal Arts, Ritsumeikan University, Ibaraki, Osaka, Japan, **4** Center for Information and Neural Networks, Advanced ICT Research Institute, National Institute of Information and Communications Technology, Suita, Osaka, Japan, **5** Graduate School of Frontiers Biosciences, Osaka University, Suita, Osaka, Japan, **6** College of Health and Life Sciences, Aston University, Birmingham, United Kingdom

⚬ These authors contributed equally to this work.
* yaman@fc.ritsumei.ac.jp

## Abstract

Short-term study-abroad (STSA) programs provide a more accessible alternative for students who would otherwise not consider engaging in academic activities overseas. Though improvements in the levels of intercultural sensitivity and general academic aspects attained by STSA programs have been previously examined, much less is known regarding the impact such programs have in the mood of students. Here, we examined changes in mood state associated with participation in an STSA program in a group of Japanese university students. Mood states were assessed using the Profile of Mood States (POMS), the Satisfaction With Life Scale (SWLS), and the Gratitude Questionnaire (GQ-6). Results indicated that the POMS mean scores of Vigor-Activity and SWLS peaked at the time immediately following participation in the STSA program; moreover, the same scores were found to be at comparable levels even one month after the end of the program. These results indicate that participation in STSA programs can positively influence the mood state of university students, suggesting that the benefits associated with participation in such programs extend beyond typically reported improvements in the academic domain.

## Introduction

Though the multifaceted benefits of participation in study-abroad programs are well documented [1], the percentage of Japanese students engaging in international academic programs remains low at approximately 1% [2]. Results from a survey sponsored by Japan's Ministry of Education, Culture, Sports, Science and Technology conducted in 2017 [3] revealed that parents who enjoy a higher socioeconomic status (i.e., income and level of education) are more likely to consider the possibility of sending their children to study abroad (55%) than lower socioeconomic status counterparts (26%), suggesting that for Japanese students, access to financial resources can act as a primary deterrent for participation in SA programs. To

author -N/A The full name of each funder -Ritsumeikan Inamori Philosophy Research Center URL of each funder website -http://www.ritsumei. ac.jp/research/riprc/eng/ Did the sponsors or funders play any role in the study design, data collection and analysis, decision to publish, or preparation of the manuscript? -NO.

**Competing interests:** NO authors have competing interests.

counter that low take-up, universities in Japan have developed a variety of study-abroad programs [4] to increase the number of domestic students gaining international academic experience by targeting subgroups of the student population that are less likely, or more reluctant, to seek international experience during their university years. Such efforts are believed to help foster more "globalized" academic environments, which are thought to be an important prerequisite for prospective domestic and international students [5, 6]. To help students overcome the barriers that may prevent their participation in short-term study abroad (STSA) programs, universities have resorted to different strategies, such as minimizing language requirements or offering access to financial support. (According to the nomenclature guideline provided by Japan's Ministry of Education, Culture, Sports, Science and Technology, programs that last from 1 week to three months should be denominated 'short-term', whereas programs that last from 6 months to two years should be denominated 'long-term' (https://tobitate.mext.go.jp/univ/planguide/)).

It is generally reported that long-term study-abroad programs, lasting one semester or more, bring greater benefits than STSA programs [7]. However, because STSA programs provide a more accessible and affordable alternative than long-term programs, they should not be overlooked, as they target a subgroup of the student population that would otherwise not engage in any type of international academic activity. Chieffo et al. [8] have noted that most of the more recent study abroad (SA) programs have a short-term character. As a result of this trend, most universities in Japan now offer STSA programs as part of their official curricula. Contrarily to the belief that programs of longer duration always generate better outcomes, a few studies have provided evidence that that is not necessarily the case. For instance, a recent study examined the effects of length of stay (LoS) in the pragmatic competence of Chinese students who underwent SA programs of three different durations, i.e., 9 months, 20 months, and 38 months [9]. Results revealed that the number of indirect request strategies produced by the SA students (a measure of pragmatic competence) that were highly similar to those produced by native speakers was not related to the LoS, indicating that even in the domain of second-language acquisition, outcomes do not necessarily intensify with longer LoS.

The benefits associated with SA programs have been measured in various ways, including both intercultural sensitivity [6] and desire to attend graduate school [7], which are relevant aspects in the context of international education. Most naturally, SA programs have historically been an intense object of study in the field of applied linguistics. That is primarily because experiencing a foreign language being spoken by native speakers in their socio-cultural settings or witnessing English being used as a lingua franca in real-life contexts amounts to an 'immersion-like' learning experience [10] that is expected to be much more effective than classroom-based approaches for second-language acquisition. Sanz et al. [10] and Freed [11] comprehensively explored these issues, focusing not only on purely linguistic factors, such as language accuracy (proficiency), syntactic and semantic processing, but also on broader aspects of communicating in a foreign language, such as sociolinguistic competence, pragmatic competence, and discourse strategy, which clearly benefit more saliently from participation in SA programs.

Assessing the long-term impact of international study, Yokota et al. [12] reported that students who participated in SA programs, compared with those who did not, adapted better to new and different cultural environments. Notably, this positive outcome was associated not only with advanced language or technical skills but also with fundamental competences of emotional intelligence and socialization, leaving students with an overall progressive outlook on life.

Other studies have shown positive effects in how participants of STSA programs perceive themselves in a globalized context after obtaining international experience. Using large-scale

surveys, Chieffo et al. [13] reported that students who took part in a five-week STSA exchange program had higher levels of self-assessed "global awareness" when compared with students who only took classes about intercultural understanding in their country of origin. Kurt et al. [14] assessed notions associated with global consciousness among students undertaking a three- to four-week STSA program, at three time-points, before, during and after the program. Results indicated that a student's self-assessed acquired practical knowledge about the world was significantly enhanced upon completion of the program.

Few studies have examined the value of more time-limited exchange programs, such as those lasting only one week [7, 15], and how such programs compare with those of longer duration. However, given the poor take-up of international study by Japanese students [16], and the increasing costs of study-abroad programs, a full investigation of the effects of time-limited STSA programs appears warranted. Moreover, though there is little doubt that SA programs can have a positive impact in improving intercultural sensitivity and helping students develop their second-language skills, it is necessary to attend to the possibility that, at the same time, there could be potential downsides resulting directly from participation in such activities.

One important question concerns how the mood of students evolve through time, both during and after their study abroad experience. For instance, it has been reported that it is not uncommon for long-term SA students to experience some form of "cultural shock" in the country of destiny [17]. In addition, long-term SA students may experience a reverse culture shock upon returning to their countries of origin [18–20], as they re-encounter peers who they perceive as unable to share the newly acquired perspectives cultivated during the time spent living in a distinct cultural and linguistic context. These phenomena have been conceptualized in the past as the U-curve [21] and W-curve [22] models of cross-cultural adaptation. However, empirical data do not always support these models [23–25]; that being so, they should be regarded as "a heuristic device to provoke discussion about acculturation", as argued by Gray et al. [26]. Indeed, the notions of "culture shock/reverse culture shock" can only explain a small fraction of the changes in mood state experienced by students who have studied abroad [26]. Directly examining how STSA can affect mood will provide new insights about this important aspect of SA programs, and help highlight potential differences that may exist between effects induced by STSA programs, compared to more traditional long-term SA programs. This will have important practical implications when preparing students who plan to participate in study abroad programs.

Overall, previous reports highlight the point that SA programs may not benefit everyone, all the time. Students may face unique hardships abroad that could negatively influence their mood in the immediate and medium terms. Therefore, it is crucial to accurately assess the extent to which the mood of SA program participants is affected by their experiences overseas. That question becomes especially relevant for SA programs of short duration, since improvements in access are likely to attract a more diverse body of students. A better understanding of how such programs can influence the mood of participants will provide a solid basis upon which STSA programs can be expanded and improved to benefit participating students more fully.

Our goal in this paper was to examine changes in the mood state of individuals participating in a one-week STSA program. We employed metrics that are typically applied in experimental psychology (see Methods for details), but which have thus far been less favored in the field of SA research [27, 28]. To examine the impact on the mood state of students participating in the one-week STSA program, data collection was performed at two time-points before and two time-points after their time abroad.

## Methods

### Participants

The STSA participants consisted of 40 Japanese undergraduate students attending a Japanese university (24 males, 16 females, mean age 19.2 years old, [range 18–22], SD = 1.0). They participated in a STSA program of one week duration, called the Global Fieldwork Project (more details in section below). Students taking part in the program were enrolled in several different majors: Letters (30%); Economics (17.5%); Sociology (15%); Life Sciences (7.5%); Policy Science (7.5%); Gastronomy Management (5%); Business Administration (5%); Science and Engineering (5%); Comprehensive Psychology (2.5%); International Relations (2.5%); and Information Science and Engineering (2.5%). A total of 42.5% of participants in the STSA program were freshmen, 40% were sophomores, 15% were juniors and 2.5% were seniors. The program had two destinations, with 65% of the students conducting the fieldwork in Penang, Malaysia, and 35% conducting the fieldwork in Phnom Pen, Cambodia. Students visited the respective countries for the first time for the occasion of this STSA program. All participants had basic English speaking and writing ability but were not able to communicate in other languages spoken locally. This study was approved by the ethics and safety committee of the National Institute of Information and Communications Technology. All students provided written informed consent prior to participation in the study.

### NEO-Five Factor Inventory (NEO-FFI)

To ensure that participants of the STSA program were a representative student sample, we asked participants to rate items in the NEO Five-Factor Inventory (NEO-FFI, [29]), and compared them with the scores given by a group of students from the same university who neither participated in the program nor spent any time overseas during the same period. Personality traits are thought to partially explain individual differences regarding emotion processes and mood (e.g., [30, 31]). Moreover, because participants in the SA program were self-selected, it was important to verify that this cohort matched well a sample of students from the same university who did not take part in the SA program. Ensuring that there were no significant inter-group differences regarding a well verified model of personality traits would minimize the chances that study results were trivially rooted on spurious confounds.

There were 38 undergraduate students (16 males, 22 females, mean age 19.4 years old, range [18–22], SD = 0.9) in the comparison group (25.6% freshmen, 74.4% sophomores). Participants in the comparison group were from the College of Life Sciences (35.9%) and the College of Pharmaceutical Sciences (46.1%). The NEO-FFI is a shortened version of the NEO Personality Inventory-Revised, an implementation of an empirically validated five-factor model of human personality [32]. The FFI is one of the most extensively applied models of personality currently in use [33], describing individual differences in terms of five personality traits: Neuroticism, Extraversion, Openness, Agreeableness, and Conscientiousness. This personality model has been examined using both cross-sectional and longitudinal studies on populations of different ages and cultural backgrounds [34–36]. The internal consistency (Cronbach's alpha) of the NEO-FFI items used to compute the scores in the 5 dimensions, based on data from a cohort of 659 Japanese adults (range [21–87] years old), was found to be mostly in the reliable range (Neuroticism, alpha = .83, Extraversion, alpha = 0.78, Openness, alpha = .75, Agreeableness, alpha = .68, Conscientiousness, alpha = .77). The one-week test-retest reliability of the NEO-PI-R, a superset of the NEO-FFI, based on data from a different cohort of 120 Japanese university students (range [18–22] years old), was found to be in the high range, with values between r = .84 (Neuroticism) to r = .91 (Agreeableness) [37].

## Profile of Mood States (POMS)

The mood states of participants were assessed using a Japanese translation of the Profile of Mood States (Second Edition, Adult Short Form, POMS 2®, Kanekoshobo Inc., Tokyo, Japan). The POMS was designed to assess the mood states of individuals aged 13 years and older [38]. The POMS is composed of six scales: Anger-Hostility (AH), Confusion-Bewilderment (CB), Depression-Dejection (DD), Fatigue-Inertia (FI), Tension-Anxiety (TA), and Vigor-Activity (VA). Higher POMS scores indicate greater levels of the corresponding construct. Note that only the VA scale has a positive valence, while all other scales have a negative connotation. The full version of the POMS consists of a collection of 65 self-rating items that are used to assess the respondents' feelings and emotions (e.g., "Worn-out", "Active"), allowing for a quick assessment of the respondent's mood state. Items are scored from 0 ("Not at all") to 4 ("Extremely"). Experimenters need to clearly specify the time period respondents should consider when evaluating the items (e.g., "Past week, including today" (employed in the current study), "Right now"). Here, we used the short version of the POMS which consists of a select 30-item subset of the full version.

An aggregate score of total mood disturbance (TMD) can be computed based on the 6 POMS scores, with greater scores indicating higher disturbance. The POMS individual scales, or the TMD score, have been employed to monitor natural changes in mood state or alterations in mood state following behavioral interventions in clinical, athletic and psychology research settings [39–41]. A few studies have employed the POMS in researching study-abroad programs (e.g. [42]). The internal consistency (Cronbach's alpha) of the 6 scales, based on the responses from 397 Japanese respondents of similar age (18–29 years old) was found to be in the reliable range ([.62-.95]). The one-week test-retest reliability based on data from 22 Japanese respondents of similar age (20–29 years old) was found to be in the high range (r values in the range [.767-.915]) [43].

## Satisfaction With Life Scale (SWLS)

The Satisfaction With Life Scale (SWLS; [44]) consists of five items developed to measures an individual's satisfaction with life as a whole, regardless and beyond specific domains such as health and finances. Satisfaction with life is thought to be a fundamental component of the construct of hedonic subjective well-being [45]. The SWLS has been extensively applied to a variety of clinical and non-clinical populations across different cultural contexts [46, 47]. The two-month test-retest coefficient based on data from 76 students was found to be in the high range (r = 0.82) [45]. The 4-week test-retest correlation coefficient based on data collected from 49 Japanese participants was in the high range (r = .80). Internal consistency of the SWLS items based on the same sample set was found to be in the reliable range (alpha = .84) [48].

## Gratitude Questionnaire (GQ-6)

The Gratitude Questionnaire (GQ-6) is a six-item scale used to assess the individual disposition of experiencing the emotion of gratitude, conceptualized as an affective trait reflecting one's tendency to attend and respond to positive outcomes enabled by the actions of external actors [49]. GQ-6 scores have been found to be positively correlated with SWLS scores [50], as well as subjective scores of happiness [51] and job satisfaction [52]. The GQ-6 (Japanese version) was found to have good internal consistency reliability (alpha = .92) and good four-week test-retest reliability (r = .86), based on the data from 409 Japanese college students [53].

## Global Fieldwork Project (GFP)

The GFP is a project-oriented, fieldwork-based program, structured such that students first explore possible themes of interest and concerns regarding the country they are about to visit, and then pursue those themes with the assistance of local university students ("buddies") during the STSA period. STSA participants were recruited via an official website maintained by the university for the purpose of disseminating information about SA programs. Students obtained 2 study credits for their participation in the STSA program reported in this study. There was no English proficiency requirement (e.g., minimum TOEIC score). Applicants were required to submit an essay in Japanese about their SA goals and aspirations, which was screened by the university's selection committee. All participants received partial financial support from an alumni association to help cover the costs involved. The STSA program began with a classroom orientation session at the end of June 2018, around the time when the spring term usually ends and the summer vacation begins.

In the initial orientation session, students were required to form groups of three to discuss possible topics of study and devise a fieldwork plan to execute during the STSA. The topics discussed were quite diverse, reflecting the variety of interests and backgrounds of the participating students. Students were encouraged to form hypotheses about various socioeconomic aspects of the country they were about to visit. Self-report data indicated that the STSA participants had none or very little experience overseas.

Approximately one month after the orientation session, students traveled to Malaysia or Cambodia (the two destinations of the STSA) led by a faculty member whose role was to oversee the entire program and help the students when necessary.

During the study abroad period, their work consisted of gathering first-hand information through questionnaires and interviews with local people to test the hypotheses proposed during the orientation session. The interviews were conducted in collaboration with their "buddies". Apart from the group-based fieldwork, the program included an introductory lecture by a local university professor, an orientation session, and a farewell party. Excluding travel time to and from the overseas destination, the total period spent abroad was six days.

Two to four weeks after their return from overseas, students were required to present their findings in front of an invited audience of fellow students and academics. As part of their presentation, they reported on whether their initial hypotheses about the socioeconomic conditions of the visited country had been confirmed or not. Through personal interviews, students also reported about their overall experience taking part in the GFP, and various other aspects regarding their experience during the STSA period.

Students in the comparison group were recruited among volunteers who took courses taught by the first author (TY) at the same university; only students with no previous SA experience were included.

## Study design

During the orientation day, approximately one month prior to the scheduled departure date (Time 1, T1), participants were requested to answer the items in the POMS, SWLS, GQ-6 and NEO-FFI. To minimize demand characteristics, participants were requested to answer the items as part of their normal classroom activities. The same was true for the comparison group when completing the NEO-FFI.

Because the current study spanned no more than three months, the NEO-FFI was completed only once. The POMS, SWLS and GQ-6 were completed four times in total: (i) during the orientation day; (ii) one day before departure (T2, mid-August for the students heading to Cambodia; early September for the students heading to Malaysia); (iii) the day they arrived

back in Japan (T3, end of August for students that visited Cambodia; mid-September for students that visited Malaysia); and (iv) during the final presentation session, held over a single weekend at the end of September (T4).

## Statistical analysis

To detect differences between groups regarding personality traits, we employed a two-way, repeated measures analysis of variance (rm-ANOVA) with group (STSA and comparison groups) as a between-subjects factor, and the NEO-FFI traits (Neuroticism, Extraversion, Openness, Agreeableness, and Conscientiousness) as within-subject factors. To detect changes in the mood of STSA participants, we employed a one-way rm-ANOVA to examine the individual components of the POMS mood state (Anger-Hostility, Confusion-Bewilderment, Depression-Dejection, Fatigue-Inertia, Tension-Anxiety, and Vigor-Activity), with time (T1, T2, T3, T4) as a within-subject factor. A similar analysis was performed using the data from the SWLS and GQ6. Degrees of freedom were adjusted using Greenhouse-Geisser estimates of sphericity (Mauchly's sphericity test). All analyses were performed using SPSS version 25 (IBM, New York, USA).

## Results

### NEO-FFI trait differences between study-abroad and comparison groups

We first examined whether our study-abroad group differed from the comparison group on the five NEO-FFI traits. Results provided confirmatory evidence that there were no latent personality trait differences between the SA and comparison groups. A two-way repeated measures ANOVA revealed a significant main effect for NEO-FFI traits ($F(2.289, 173.974) = 4.327$, $p = 0.011$, Greenhouse-Geisser corrected), but no main effect of group ($F(1, 76) = 2.442$, $p = 0.122$). Moreover, there was no interaction between the NEO-FFI traits and group ($F(2.289, 173.974) = 0.2401$), $p = 0.816$). Post-hoc tests using Bonferroni correction revealed that the mean score for the NEO-FFI Extraversion trait (M = 25.78, SD = 8.63) and Agreeableness trait (M = 30.24, SD = 5.87) differed significantly ($p < 0.001$).

### POMS

**Vigor-Activity (VA) scores across time.**   Vigor-Activity scores changed significantly as a result of SA participation. We entered the scores of the component VA in a one-way rm-ANOVA, with time (T1, T2, T3, T4) as a within-subjects factor. The results showed a significant main effect of time ($F(3, 117) = 24.078$, $p < 0.0001$). Post-hoc tests using Bonferroni correction for multiple comparisons indicated that the mean VA score at time T3 (M = 13.67, SD = 4.621), collected the day after participants returned from abroad, was larger than the mean score at both time T1 (M = 9.08, SD = 4.891), T2 (M = 8.98, SD = 4.452) and T4 (M = 11.53, SD = 3.883). Furthermore, the mean score at T4 was larger than the scores collected at T1 and T2, and smaller than the score at T3. These results are shown in Fig 1A.

**Confusion-Bewilderment (CB), Fatigue-Inertia (FI) and Tension-Anxiety (TA) scores across time.**   Mean scores of the CB component were entered in a one-way rm-ANOVA, with time (T1, T2, T3, T4) as a within-subjects factor. The results showed a significant main effect of time ($F(3, 117) = 4.083$, $p < 0.013$). Post-hoc tests using Bonferroni correction indicated that the mean CB score at time T1 (M = 6.80, SD = 4.334), collected at the initial orientation session, was larger than the mean score at both time T2 (M = 5.05, SD = 3.721) and T3 (M = 5.22, SD = 3.899). These results are shown in Fig 1B.

Similarly, Fig 1C shows that there was a significant main effect of time on the FI component ($F(3, 117) = 2.730$, $p < 0.047$), with post-hoc tests using Bonferroni correction indicating that

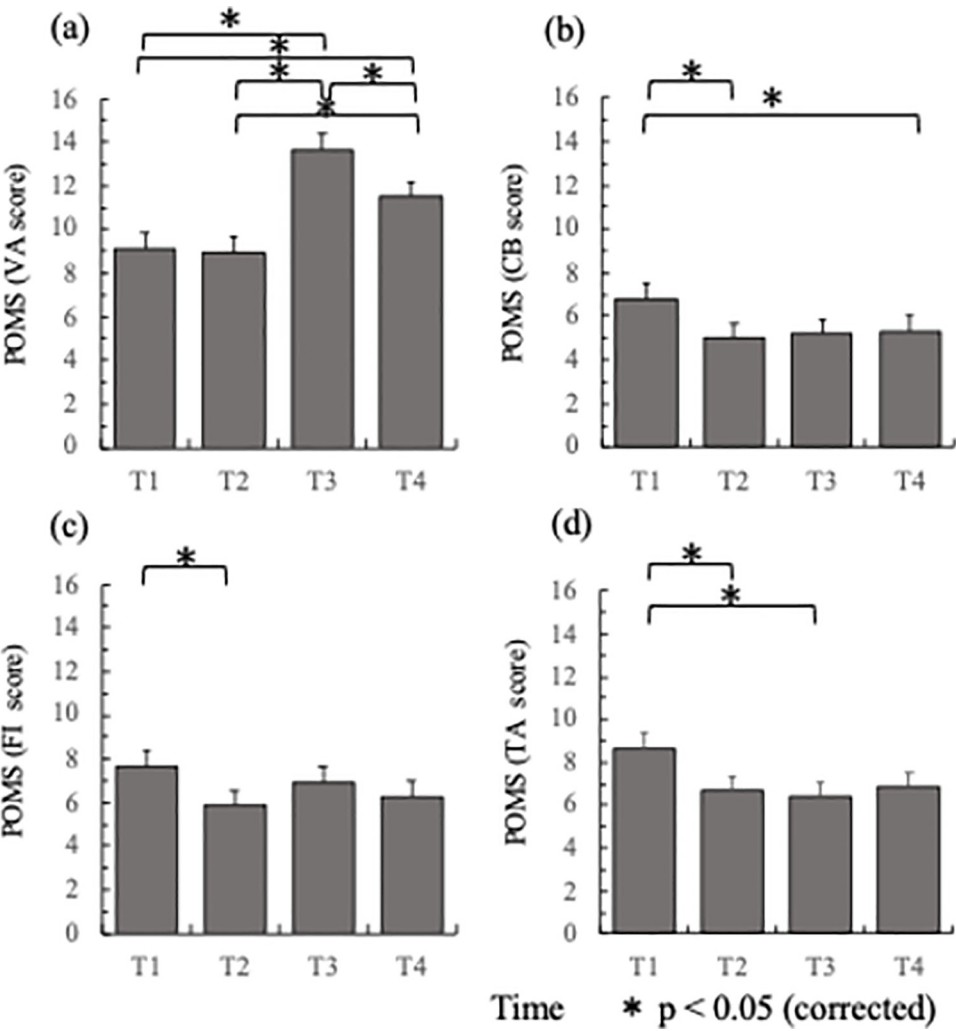

**Fig 1.** POMS scores (averages) across time. (a) Vigor-Activity (VA), (b) Confusion-Bewilderment (CB), (c) Fatigue-Inertia (FI), and (d) Tension-Anxiety (TA). T1: one month before departure overseas, T2: one day before departure, T3: the day of arrival back in Japan, and T4: 2–4 weeks after their return. The asterisk (*) indicates statistically significant pairwise differences (p < 0.05 level), corrected for multiple comparisons using the Bonferroni method. Error bars indicate one standard error.

the mean FI score at time T1 (M = 7.67, SD = 4.281) was larger than the mean score at time T2 (M = 5.90, SD = 3.875).

There was also a significant main effect of time on the mean scores for the TA component (F(3, 117) = 4.954, p < 0.003). Post-hoc tests using Bonferroni correction indicated that the mean TA score at time T1 (M = 8.65, SD = 4.554) was larger than the mean score at both time T2 (M = 6.68, SD = 4.129) and T3 (M = 6.43, SD = 3.974). These results are shown in Fig 1D.

**Anger-Hostility (AH) and Depression-Dejection (DD) scores across time.** There was no significant difference between the mean scores across time on either the AH component (F(3, 117) = 2.505, p = 0.075) or the DD component (F(3, 117) = 1.128, p = 0.302).

## Satisfaction With Life Scale (SWLS) scores across time

Mean SWLS scores were entered in a one-way rm-ANOVA, with time (T1, T2, T3, T4) as a within-subjects factor. The results showed a significant main effect of time (F(3, 117) = 12.201,

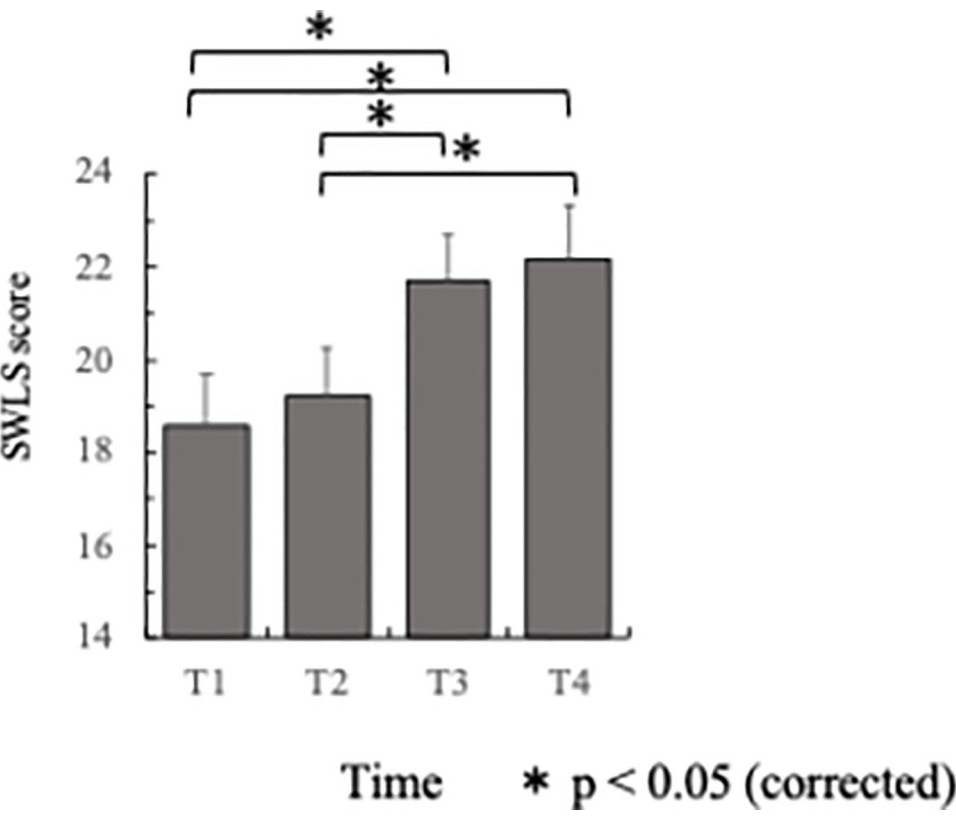

**Fig 2. SWLS scores (averages) across time at T1, T2, T3 and T4 (see caption for Fig 1 for details).** The asterisk (*)
indicates statistically significant pairwise differences (p < 0.05 level), corrected for multiple comparisons using the
Bonferroni method. Error bars indicate one standard error.

p < 0.001). Post-hoc tests revealed that the mean SWLS at T3 (M = 21.70, SD = 6.313) was
greater than the mean score at both T1 (M = 18.60, SD = 6.997) and T2 (M = 19.25,
SD = 6.392). Also, the mean score at T4 (M = 22.17, SD = 7.243) was greater than the scores at
both T1 and T2. These results are shown pictorially in Fig 2.

## Gratitude Questionnaire (GQ6) scores across time

A one-way rm-ANOVA, with time (T1, T2, T3, T4) as a within-subjects factor, revealed a sig-
nificant main effect for time on the mean scores on the Gratitude Questionnaire (GQ6) (F(3,
117) = 3.482, p < 0.018). However, post-hoc tests using Bonferroni correction for multiple
comparisons failed to detect pairwise differences between samples collected at different time-
points.

## Personal interviews (qualitative data)

We conducted semi-structured interviews with all participants after they returned to Japan.
Questions included in the interview were (1) "Please tell us some of your experiences during
the stay overseas that you were grateful for", and (2) "Please tell us what you would like to tell
other students about your STSA experience". Most participants reported having a much better
experience than they had originally expected, e.g., some participants admitted having initially
joined the STSA program mainly for the purpose of obtaining credits but realizing afterwards
that the SA period was an invaluable life experience. Moreover, most participants reported that

they would recommend the STSA program to fellow students, admittedly not for improving language skills but rather, e.g., "for the extraordinary experience that can be obtained through the program". Though we were not able to detect increases in GQ-6 scores after STSA participation, most students reported feeling more grateful for the life circumstances they enjoyed in Japan, e.g., for being able to attend a private university, for being supported by their parents, for having the support of a network of good friends. All in all, these qualitative results provide additional evidence of the positive impact STSA participation can have on university students.

## Discussion

In this study, we sought to examine changes in the mood of Japanese university students who took part in a short-term study-abroad (STSA) program of one week duration, in which fieldwork activities were undertaken with the help of local students (in either Cambodia or Malaysia). We first provided confirmatory evidence that students taking part in the program were a representative sample of the student population, using the NEO-Five Factor Inventory to establish that there were no significant differences in major personality traits between study-abroad participants and a comparison group drawn from across the university sector. We next employed POMS to assess the mood states of the study-abroad group, as defined by the categories Anger-Hostility, Confusion-Bewilderment, Depression-Dejection, Fatigue-Inertia, Tension-Anxiety and Vigor-Activity. Our results show that Vigor-Activity increased significantly during their stay overseas (T3), when compared with the levels observed before their departure (T2) (see Fig 1A), and generally remained high throughout the following weeks (T4).

Results from a study by Wen-Lin and Gregory (2015) showed that individual scores of the Intercultural Effectiveness Scale (IES) [54], which measures overall cultural competence and intercultural attitudes, were positively correlated with the Big-Five Personality Inventory factors [55]. More specifically, they showed that six IES factors, namely, behavioral flexibility, interaction relaxation, interactant respect, message skills, identity maintenance, and interaction management, were positively correlated with four personality factors (extraversion, agreeableness, conscientiousness, openness). In addition, the same study showed that the six IES factors were negatively correlated with the neuroticism factor. The absence of intergroup differences regarding personality traits in our sample would suggest that, albeit indirectly, it is not likely that there were pre-existing differences of intercultural effectiveness between SA participants and individuals in the comparison group that could explain these findings.

Other studies have reported associations between the Big Five personality dimensions (i.e., agreeableness and openness) and intent to study abroad (e.g., Nieoff et al. [56]), as well as associations between the Big Five personality dimensions (i.e., extraversion agreeableness and neuroticism) and the desire of expatriates to continue working abroad and fulfill their terms as originally planned (as opposed to terminating their terms earlier to return to their homelands) [57]. These past studies would suggest that there should be differences in the NEO-FFI scores between the SA participants and individuals in the comparison group; however, that was not the case in the current sample. We believe this discrepancy may be explained by differences in the specific circumstances revolving each one of the studies. One salient difference concerns the length of stay between the study-abroad programs. Nieoff et al. [56] examined students who were going to join a semester-long study-abroad program. In comparison, participants in the current study only committed to a one-week study-abroad program. This arguably made the STSA much more accessible to participants, i.e., the STSA program sampled from a much larger body of students than typical long-term SA programs, which resulted in SA participants being much more similar to matched individuals in the comparison group than in previous reports.

Caligiuri [57] examined the relationship between personality characteristics and the aspirations of a group of expatriates and found that the Big Five personality dimensions were associated with the will to continue living abroad. However, in their study, participants had lived in the United States for 1.8 years on average; more importantly, the mean age of participants in their sample was 40 years old. In contrast, participants in the current study were all university students aged from 18 to 22 years old, departing to a STSA program of only one week. Overall, we believe these fundamental differences make it very difficult to directly compare these studies, and likewise, apply the results from one study to predict the results of another.

Mood states, as measured by scales such as the POMS, have been shown to be enhanced using a variety of different manipulations, including meditation [58, 59], sitting isometric yoga [60], massage therapy [61], exposure to nature [62] and physical activities [63]. Here, we attribute the observed enhancements in the scores of Vigor-Activity to participation in the study-abroad program. In addition, using the Satisfaction With Life Scale (SWLS), we showed that participation in STSA programs, even when limited to one week duration, resulted in a significant improvement in a core aspect of subjective well-being that was sustained for at least a few weeks after the students returned from overseas (see Fig 2). On the other hand, contrary to our expectations, no differences in GQ-6 scores were detected.

We conclude that STSA programs limited in duration to just one week are capable of positively impacting the affective state of students. This is of great importance when considering that short-term programs are a much more accessible alternative compared to long-term programs. As such, STSA programs are likely to provide a more feasible entry point for students who otherwise would not consider the possibility of obtaining overseas experience during their years in college. Continued validation of STSA programs using scales such as the BEVI (Beliefs, Events, Values Inventory), IDI (Intercultural Development Inventory) and GPI (Global Perspectives Inventory) (reviewed in [64]) will enable educators to construct more accurate expectations of the long-term benefits of participating in such programs.

It is important to attend to the fact that presently there are only a few studies in the literature that have specifically looked at SA programs of such a short duration. Indeed, the benefits of STSA may be limited in several respects. For example, Suzuki et al. [65] reported that the foreign language speaking proficiency of students who participated in STSA programs for about three weeks was only "slightly improved". Thus, it possible and likely that improvements in language proficiency are even less prominent, if at all, in SA programs of much shorter durations such as one week. In addition, Tarchi et al. [66] showed that there were virtually no changes in the development of intercultural competence after 3 months studying abroad, suggesting that a one-week SA is likely to have very little impact in that respect. Moreover, to more accurately assess the effective magnitude of the changes in mood observed among STSA students, future studies should track participants over longer periods of time (e.g., one year) to compare variations in affective state relative to other significant events in the school calendar (e.g., exam and holiday periods).

How do the enhancements in Vigor-Activity that were observed among the STSA students relate to the changes in mood experienced by long-term SA students? Here, SA students felt energized upon return, as indicated by the increases in Vigor-Activity (VA) scores. Furthermore, even though the VA scores decreased after one month, they were still significantly higher than the levels observed at the onset of the study. These results, combined with the fact that no significant effects were detected for the negative affect dimensions (e.g., POMS Depression-Dejection scores), and that the well-being scores (measured by SWLS) improved during STSA and remained at similar levels one month after, suggest that the SA participants did not suffer a reverse culture shock in the aftermath of the STSA, in contrast with what has been reported about long-term SA program participants. More importantly, the current results

indicate that mood changes for short-term SA students may have peculiar and distinctive characteristics that need to be investigated in more details in future research.

Overall, the psychological impact of studying abroad must be considered from different perspectives to be appreciated in its entirety. As Gray et al. have pointed out, when considering adaptation and adjustment during reentry, the strength of personal factors and situational factors must be considered in a holistic manner. Cultural distance (i.e., differences between one's own culture and the culture in the local of destination) and the degree of immersiveness in the host country's culture are also thought to cause difficulties in re-entry adjustment [26]. The current findings showing that the STSA program had little (negative) effect upon reentry may provide new insights for understanding why negative effects of reentry occur in long-term SA. A clear understanding of mood effects elicited by participation in SA programs will guide the development of educational content for prospective participants, to better orient them about the potential pitfalls in advance, and help them put in effect strategies to effectively cope with changes in mood state upon return.

## Supporting information

**S1 Data.**
(XLSX)

## Author Contributions

**Conceptualization:** Tsukasa Yamanaka.

**Data curation:** Noriko Yamagishi, Norberto Eiji Nawa.

**Funding acquisition:** Tsukasa Yamanaka.

**Investigation:** Tsukasa Yamanaka.

**Methodology:** Noriko Yamagishi.

**Project administration:** Tsukasa Yamanaka.

**Resources:** Tsukasa Yamanaka.

**Supervision:** Tsukasa Yamanaka.

**Visualization:** Noriko Yamagishi, Norberto Eiji Nawa.

**Writing – original draft:** Tsukasa Yamanaka.

**Writing – review & editing:** Noriko Yamagishi, Norberto Eiji Nawa, Stephen J. Anderson.

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
