## [Decision Letter · Decision Letter 0]

28 May 2021

PONE-D-21-12484

Assessing changes in mood state in university students following short-term study abroad

PLOS ONE

Dear Dr. Yamanaka

Thank you for submitting your manuscript to PLOS ONE. After careful consideration, we feel that it has merit but does not fully meet PLOS ONE’s publication criteria as it currently stands. Therefore, we invite you to submit a revised version of the manuscript that addresses the points raised during the review process.

You should address all the suggestions made by Reviewer 1, in particular those regarding the method and discussion sections.

We look forward to receiving your revised manuscript.

Kind regards,

Berta Schnettler

Academic Editor

PLOS ONE

Journal Requirements:

Reviewers' comments:

Reviewer's Responses to Questions

**Comments to the Author**

1. Is the manuscript technically sound, and do the data support the conclusions?

Reviewer #1: Partly

Reviewer #2: Yes

Reviewer #3: Yes

2. Has the statistical analysis been performed appropriately and rigorously? 

Reviewer #1: Yes

Reviewer #2: Yes

Reviewer #3: Yes

3. Have the authors made all data underlying the findings in their manuscript fully available?

Reviewer #1: Yes

Reviewer #2: Yes

Reviewer #3: Yes

4. Is the manuscript presented in an intelligible fashion and written in standard English?

Reviewer #1: Yes

Reviewer #2: Yes

Reviewer #3: Yes

5. Review Comments to the Author

Reviewer #1: PONE-D-21-12484

This manuscript explores a topic of increasing interest in the study abroad literature, the outcomes of short-term study abroad (STSA) programs. Few well-designed empirical studies and even fewer with participants from outside North American and Europe have addressed this issue. However, I believe several concerns would need to be addressed prior to publication, most importantly dealing with the rationale for measuring mood. These are detailed below.

Introduction

• The authors state that alternative study abroad (SA) programs in Japan have been developed to target subgroups of students less likely to participate in traditional SA. Could the authors elaborate on the nature of these subgroups? For example, in the literature from North American, the students underrepresented in SA tend to be first generation students and Student of Color.

• A critical missing component of the Introduction is some discussion of the value of investigating mood as an outcome of STSA (other than the fact that it has not been done before). What is the purpose of demonstrating changes in these affective states and what would study abroad researchers or practitioners do with this knowledge?

Methods

• Regarding self -selection into the Global Fieldwork Project, please indicate how participants (those who attended the STSA and the controls) were recruited and whether there were any explicit benefits to participation in the program.

• Please explain the rationale behind testing the SA participants and controls for pre-existing differences on the NEO-FFI. Perhaps the logic is that FFM traits (such as Extraversion and Neuroticism) would be correlated with mood? Also, if the purpose of the NEO-FFI assessment is to “ensure that the participants of the STSA program were a representative student sample” and rule out differences linked to self-selection, it may be more relevant to identify pre-existing differences in indices of cultural competence or intercultural attitudes.

• For readers unfamiliar with the POMS, please provide some information on the nature of each of the subscales including, if possible, sample items. The authors state that the POMS allows for “a quick assessment of transient, fluctuating feelings, as well as enduring affective states.” Is it the case that each of the subscales utilized assess both transient and enduring affect? If so, how should we interpret the results of this assessment?

• Please provide reliability coefficients for each of the measures and evidence that any of the scales developed and standardized on Western samples are appropriate for use with Japanese students.

Results

• In terms of the NEO-FFI results, it seems odd that there are not pre-existing differences in the FFM traits that have been consistently associated with study abroad intent and participation in the literature (see, for example Caligiuri, 2000; Niehoff et al., 2017). Why might that be?

Discussion

• What should readers conclude about the finding that a bump in Vigor or subjective well-being may follow STSA. Most study abroad professionals will report that students return from their sojourn excited and energized, and then often “crash” when they learn that their friends and family have little understanding or interest in their experiences or other adjustment challenges occur. What are the implications of this finding? Is it possible to interpret the change in mood as indicating some similarity between STSA and programs of a longer duration?

References

Caligiuri, P. M. (2000). The Big Five personality characteristics as predictors of expatriate's desire to terminate the assignment and supervisor-rated performance. Personnel Psychology, 53(1), 67–88. https://doi.org/10.1111/j.1744-6570.2000.tb00194.x

Niehoff, E., Petersdotter, L., & Freund, P. A. (2017). International sojourn experience and personality development: Selection and socialization effects of studying abroad and the Big Five. Personality and Individual Differences, 112, 55- 61. https://doi.org/10.1016/j.paid.2017.02.043

Reviewer #2: The paper discusses the effect of short-term study abroad.

In introduction, the authors distinguished long-term study abroad from short term study abroad. However, it would be useful to explicitly define how long is long-term SA? How long is short term SA?

P2. Paragraph 2. The authors reviewed a few studies in different aspects. It would be good to mention some linguistic studies on SA, particularly those explicitly stating short-term SA benefits vs. long-term SA benefits. For example:

Ren, Wei. 2019. Pragmatic development of Chinese during study abroad: A cross-sectional study of learner requests. Journal of Pragmatics, 146, 137-149.

Sanz, Cristina & Alfonso Morales-Front (Eds.), The Routledge Handbook of Study Abroad Research and Practice. New York: Routledge.

Participants. What are the age ranges of the participants? Had they had SA experience before taking part in the study?

Findings. In addition to the inferential statistic results, it would be better if the authors could also provide some descriptive results.

Discussion. Since the study investigated students’ SA for only 1 week, the short term SA may lead to un-development in some aspects, for example GQ-6. The possibility of short-term SA limitation should be discussed.

Reviewer #3: The study was carried out robustly and presented in an intelligible manner. It would benefit from providing more information about the participants in terms of whether or not they had previous experience abroad or with the countries mentioned.

6. PLOS authors have the option to publish the peer review history of their article (what does this mean?). If published, this will include your full peer review and any attached files.

Reviewer #1: No

Reviewer #2: No

Reviewer #3: No

---

## [Author Response · Author response to Decision Letter 0]

16 Sep 2021

---***---***---***---***---***---***---***---***---

COMMENTS FROM/REPLIES TO REVIEWER #1

---***---***---***---***---***---***---***---***---

Reviewer #1: PONE-D-21-12484 

This manuscript explores a topic of increasing interest in the study abroad literature, the outcomes of short-term study abroad (STSA) programs. Few well-designed empirical studies and even fewer with participants from outside North American and Europe have addressed this issue. However, I believe several concerns would need to be addressed prior to publication, most importantly dealing with the rationale for measuring mood. These are detailed below. 

Introduction

• The authors state that alternative study abroad (SA) programs in Japan have been developed to target subgroups of students less likely to participate in traditional SA. Could the authors elaborate on the nature of these subgroups? For example, in the literature from North American, the students underrepresented in SA tend to be first generation students and Student of Color.

Thank you for requesting this clarification. According to a survey sponsored by Japan’s Ministry of Education, Culture, Sports, Science and Technology*, one of the main determinants of whether or not a student will have the opportunity to study abroad happens to be the Socio-Economic Status (SES) of the parents (the SES is a composite score based on family income and the parents’ level of education).

Results from that survey showed that 55% of the parents in the highest SES stratum, who have children attending elementary school or junior high school, entertain the possibility of sending their children to study abroad in the future. However, for parents in the lowest SES stratum, that number drops to 26%. This clearly suggests that whether university students in Japan will have an opportunity to pursue academic activities overseas seems to depend primarily on the economic status of their parents, an unsurprising fact given the costs involved with participating in a long-term SA program. On the other hand, because the requirements (financial and otherwise) for joining short-term SA programs are much less demanding in comparison, the decision of whether to participate in a STSA program is likely to be much less influenced by the parents’ SES. We added this new information to the Introduction.

* Unfortunately, we were not able to find an English translation of the above-mentioned survey. Details of the survey (in Japanese), conducted in 2017, can be found in the URLs below: https://www.mext.go.jp/b_menu/shingi/chousa/shotou/112/shiryo/attach/1381166.htm

https://www.nier.go.jp/17chousa/pdf/17hogosha_factorial_experiment.pdf

• A critical missing component of the Introduction is some discussion of the value of investigating mood as an outcome of STSA (other than the fact that it has not been done before). What is the purpose of demonstrating changes in these affective states and what would study abroad researchers or practitioners do with this knowledge? 

Thank you for raising this important point. We more clearly developed the rationale as for why we focused on assessing mood changes associated with a short-term SA in the Introduction. We hope the motivation behind our study is much clearer in the revised manuscript.

Methods

• Regarding self -selection into the Global Fieldwork Project, please indicate how participants (those who attended the STSA and the controls) were recruited and whether there were any explicit benefits to participation in the program.

Thank you for directing our attention to important details that were missing in the original manuscript. Study participants were recruited via an official website maintained by the university for the purpose of disseminating information about various study abroad programs. Students obtained 2 study credits for their participation in the STSA program reported in this study. There was no English proficiency requirement (e.g., minimum TOEIC score). Applicants were required to submit an essay in Japanese about their SA goals to the university, which was screened by the university’s selection committee. All participants received partial financial support provided by an alumni association to help cover the costs involved. Students in the comparison group were recruited among volunteers who took courses taught by the first author (TY) at the same university; students with study abroad experience were excluded. This information was added to the revised manuscript.

• Please explain the rationale behind testing the SA participants and controls for pre-existing differences on the NEO-FFI. Perhaps the logic is that FFM traits (such as Extraversion and Neuroticism) would be correlated with mood? Also, if the purpose of the NEO-FFI assessment is to “ensure that the participants of the STSA program were a representative student sample” and rule out differences linked to self-selection, it may be more relevant to identify pre-existing differences in indices of cultural competence or intercultural attitudes.

Thank you for asking for this clarification. Personality traits are thought explain in part many individual differences regarding emotion processes and mood (e.g., Erbas et al., 2014; Dwan et al., 2017). As you correctly inferred, one of the reasons why we tested for differences in personality traits (as measured using the widely adopted NEO-FFI dimensions) between the SA participants and matched participants in the comparison group was to ensure that there were no significant intergroup differences, at the most basic level, that could end up becoming confounds to the main experimental manipulation results. We were glad to be able to confirm that both groups did not differ with respect to their scores across the 5 NEO-FFI dimensions in any significant way. 

Another concern we had, as you correctly pointed out, was the self-selected nature of the individuals in the SA group. Could it be that the students who actively decided to pursue SA were distinct in some way, compared to non-SA students? Failing to detect significant differences concerning the NEO-FFI increased our confidence that the SA participants were indeed a representative sample of the more general student population.

 We did not collect data directly related to potential pre-existing differences regarding cultural competence or intercultural attitudes. However, results from a study by Wen-Lin and Gregory (2015) showed that individual scores of the Intercultural Effectiveness Scale (IES), a metric of overall cultural competence and intercultural attitudes (Portalla and Chen, 2010), were positive correlated with factors of the Big-Five Personality Inventory, which assess the same personality constructs as the NEO-FFI. More specifically, they showed that six IES factors, namely, behavioral flexibility, interaction relaxation, interactant respect, message skills, identity maintenance, and interaction management, were correlated positively with four factors, namely, extraversion, agreeableness, conscientiousness, openness. In addition, the same study showed that the six IES factors were negatively correlated with neuroticism. The absence of intergroup differences regarding personality traits in our sample suggests, albeit indirectly, that it is not likely that there were pre-existing differences of intercultural effectiveness between the SA participants and individuals in the comparison group. All in all, because SA participants and comparison participants were well matched for personality traits, we do not think there were significant discrepancies in terms of cultural competence and/or intercultural attitudes that could have affected our main results.

We updated the revised manuscript with the information above.

Erbas, Y., Ceulemans, E., Pe, M. L., Koval, P., and Kuppens, P.. Negative emotion differentiation: Its personality and well-being correlates and a comparison of different assessment methods. Cognition & Emotion. 28:7, 1196-1213. 2014.

Dwan, T., Ownsworth, T., Donovan, C., and Lo, A. H. Y.. Reliability of the NEO Five Factor Inventory short form for assessing personality after stroke. International Psychogeriatrics. 29:7, 1157-1168. 2017.

Portalla, T., & Chen, G.-M. The development and validation of the intercultural effectiveness scale. Intercultural Communication Studies. 2010; 19(3): 21-37.

Wen-Lin W, Gregory SC. The role of personality and intercultural effectiveness towards study abroad academic social activities. International Journal of Research Studies in Psychology. 2015;4(4): 13-27. 

• For readers unfamiliar with the POMS, please provide some information on the nature of each of the subscales including, if possible, sample items. The authors state that the POMS allows for “a quick assessment of transient, fluctuating feelings, as well as enduring affective states.” Is it the case that each of the subscales utilized assess both transient and enduring affect? If so, how should we interpret the results of this assessment?

Thank you for this suggestion. In the original manuscript, we stated that POMS can assess “transient, fluctuating feelings, as well as enduring affective states” without describing more precisely how the target periods of time are actually defined, which would of course change the meaning of “transient” and “enduring.” We corrected that imprecision in the revised manuscript. In fact, when administering the test, the experimenter has to clearly specify the period of time respondents should consider when evaluating the items in the POMS; default options are “Past week, including today” and “Right now”. The experimenter also has the choice of freely customizing the target period. That leeway allows one to test for transient states OR enduring states, though arguably not both, at the same time. 

In the current study, the target period of time respondents had to consider was “Past week, including today”. We added these clarifications in the revised version of the manuscript (Methods; POMS section).

The full POMS contains 65 items (adjectives) that describe six different dimensions of mood: Anger-Hostility (AH), Confusion-Bewilderment (CB), Depression-Dejection (DD), Fatigue-Inertia (FI), Tension-Anxiety (TA), and Vigor-Activity (VA). In this study, though, we used the short version of the POMS which has a subset of select 30 out of the 65 items, e.g., ‘(I feel) Angry 0: not at all, 4: extremely’ (one of the Anger-Hostility items), ‘(I feel) Worn out 0: not at all, 4: extremely’ (one of the Fatigue-Inertia items), ‘(I feel) Tense 0: not at all, 4: extremely’ (one of the Tension-Anxiety items). As stated above, participants were asked to evaluate the items with respect to the previous week, which was a suitable period of time to compare changes in mood before and after participation in the STSA program. 

• Please provide reliability coefficients for each of the measures and evidence that any of the scales developed and standardized on Western samples are appropriate for use with Japanese students. 

Thank you for requesting this clarification. We used the Japanese versions of the NEO-FFI, POMS, SWLS and GQ-6. The reliability coefficients (test-retest r and Cronbach’s alpha) for each one of the measures based on data collected from Japanese participants are: (1) for NEO-FFI, r = [.84 - .91], α = [.68 - .83], (2) for POMS, r = [.767 - .915], α = [.62 - .95] , (3) for SWLS, r = 0.80, α = 0.84, and (4) for GQ-6, r = 0.92 , α = 0.86.

We added the information above, together with relevant references, in the Methods section of the revised manuscript.

Results

• In terms of the NEO-FFI results, it seems odd that there are not pre-existing differences in the FFM traits that have been consistently associated with study abroad intent and participation in the literature (see, for example Caligiuri, 2000; Niehoff et al., 2017). Why might that be? 

Thank you for this comment. Although some studies have shown associations between some of the Big Five personality dimensions (i.e., agreeableness and openness) and students’ study abroad intent (e.g., Nieoff et al, 2017), and between some of the Big Five personality dimensions (i.e., extraversion, agreeableness and neuroticism) and the desire of expatriates to fulfill their terms overseas (as opposed to returning early) (Caligiuri, 2000), we were not able to detect differences regarding the Big Five personality dimensions between the STSA and the comparison groups. 

This discrepancy may be explained by differences in length of the study abroad programs involved. Nieoff et al. (2017) examined students who were going to join a semester-long study abroad. In comparison, the STSA program in the current study lasted only one week. This arguably made the STSA more accessible to a larger cohort of students, which in turn resulted in the fact that SA participants were highly similar to students in the comparison group (as indicated by the lack of significant differences vis-a-vis personality traits between the two groups). Caligiuri (2000) examined the relationship between personality characteristics and the aspirations of a group of expatriates under a framework based on the theory of evolutionary personality psychology. In their study, participants had lived in the United States for 1.8 years on average; more importantly, the mean age of participants in their sample was 40 years old. In contrast, in our study, participants were all university students aged from 18 to 22 years old; moreover, the duration of the STSA was only one week. We believe these differences make it very difficult to directly compare these studies, and likewise, apply the results from one study to predict the results of another. We added these arguments to the Discussion section.

Caligiuri, P. M. (2000). The Big Five personality characteristics as predictors of expatriate's desire to terminate the assignment and supervisor-rated performance. Personnel Psychology, 53(1), 67–88. https://doi.org/10.1111/j.1744- 6570.2000.tb00194.x 

Niehoff, E., Petersdotter, L., & Freund, P. A. (2017). International sojourn experience and personality development: Selection and socialization effects of studying abroad and the Big Five. Personality and Individual Differences, 112, 55- 61. https://doi.org/10.1016/j.paid.2017.02.043

Discussion

• What should readers conclude about the finding that a bump in Vigor or subjective well-being may follow STSA. Most study abroad professionals will report that students return from their sojourn excited and energized, and then often “crash” when they learn that their friends and family have little understanding or interest in their experiences or other adjustment challenges occur. What are the implications of this finding? Is it possible to interpret the change in mood as indicating some similarity between STSA and programs of a longer duration? 

References

Caligiuri, P. M. (2000). The Big Five personality characteristics as predictors of expatriate's desire to terminate the assignment and supervisor-rated performance. Personnel Psychology, 53(1), 67–88. https://doi.org/10.1111/j.1744- 6570.2000.tb00194.x 

Niehoff, E., Petersdotter, L., & Freund, P. A. (2017). International sojourn experience and personality development: Selection and socialization effects of studying abroad and the Big Five. Personality and Individual Differences, 112, 55- 61. https://doi.org/10.1016/j.paid.2017.02.043

Thank you for providing us with this insight. This is an issue at the very core of the motivation underlying the current study, namely, the need to address the methodological gap regarding how the mood of STSA participants change during and after a period of time in a distinct cultural and linguistic setting. 

 In the Introduction of the revised manuscript, we allude to Lysgaard's U-curve Adjustment Hypothesis theory, which distinguishes 4 different periods that (long-term) SA students may go through during the SA period. Crucially, what the current results suggest is that the experience of STSA students seems to be quite different to what long-term students normally undergo; our data indicate that after a (short) period overseas, students come back energized (as indicated by higher levels of Vigor-Activity). Furthermore, even after one month, the VA scores are still higher than the levels observed at the onset of the study. 

Even though the current results still need to be independently replicated before broader generalizations are made, we believe that, most importantly, they indicate that changes in mood experienced by STSA students may have unique and distinct characteristics that need and deserve to be investigated in their own right. 

---***---***---***---***---***---***---***---***---

COMMENTS FROM/REPLIES TO REVIEWER #2

---***---***---***---***---***---***---***---***---

Reviewer #2: The paper discusses the effect of short-term study abroad.

In introduction, the authors distinguished long-term study abroad from short term study abroad. However, it would be useful to explicitly define how long is long-term SA? How long is short term SA? 

Thank you for requesting this clarification. According to the nomenclature guideline provided by Japan’s Ministry of Education, Culture, Sports, Science and Technology*, (1) programs that last from 1 week to three months should be denominated ‘short-term’, whereas (2) programs that last from 6 months to two years should be denominated ‘long-term’.

*Site about studying overseas from the Japan’s Ministry of Education, Culture, Sports, Science and Technology (in Japanese):

https://tobitate.mext.go.jp/univ/planguide/

P2. Paragraph 2. The authors reviewed a few studies in different aspects. It would be good to mention some linguistic studies on SA, particularly those explicitly stating short-term SA benefits vs. long-term SA benefits. For example: Ren, Wei. 2019. Pragmatic development of Chinese during study abroad: A cross-sectional study of learner requests. Journal of Pragmatics, 146, 137-149. 

Sanz, Cristina & Alfonso Morales-Front (Eds.), The Routledge Handbook of Study Abroad Research and Practice. New York: Routledge. 

Thank you very much for this suggestion. We now refer to SA linguistic studies in the Introduction of the revised manuscript, with some references (including the ones you suggested).

Participants. What are the age ranges of the participants? Had they had SA experience before taking part in the study? 

Thank you very much for this inquiry. For some reason, this very important piece of information was missing in the original manuscript. The mean age of the SA participants was 19.2 years old (SD = 1.0, range [18-22]), whereas the mean age of the students in the comparison group was 19.4 years old (SD = 0.9, range [18-22]). SA participants visited the respective countries for the first time at the occasion of the STSA. Students had basic English speaking and writing skills but were not able to communicate in other languages spoken locally. Self-reports indicated that participants had none or very little experience overseas. We added all the above information in the Methods section.

Findings. In addition to the inferential statistic results, it would be better if the authors could also provide some descriptive results. 

Thank you very much for this suggestion. After much deliberation, we decided not to include additional descriptive statistics regarding the collected measures (other than the already included means and SDs) because we thought the added details would not necessarily lead to a better understanding of the results of the current study. Rather, we opted to add a qualitative description of the reports obtained from semi-structured interviews conducted with all participants after they returned to Japan. We think those reports do help provide some important insights about the potential positive impact STSA participation can have in the mood at large of university students.

Questions included in the interview were (1) “please tell us some experiences that you were grateful for during the stay”, and (2) “please tell us what you would like to tell other students about your experience overseas”. Most participants reported having a much better experience than they had originally expected, e.g., some participants admitted having joined the STSA program mainly for the purpose of obtaining credits but realizing afterwards that they had had an invaluable life experience. Moreover, most participants reported that they would recommend the STSA program to fellow students, admittedly not for improving language skills but rather, e.g., “for the extraordinary experience that can be obtained through the program”. Though we were not able to detect increases in GQ-6 scores (which measures trait gratitude orientation) after participation in the STSA, most students reported feeling more grateful for their life situations in Japan, e.g., for being able to attend a private university, for being supported by their parents, for having the support of a network of good friends. We believe these qualitative results provide additional evidence of the positive impact STSA participation can have on the mood at large of university students and shall be explored further in future studies.

This information was added in a new section in the revised manuscript.

Discussion. Since the study investigated students’ SA for only 1 week, the short term SA may lead to un-development in some aspects, for example GQ-6. The possibility of short-term SA limitation should be discussed. 

Thank you very much for this comment. We discuss some further limitations of STSA in the Discussion section, with some additional references. We hope that will provide a broader perspective of the potential benefits as well as the limitations of STSA programs.

---***---***---***---***---***---***---***---***---

COMMENTS FROM/REPLIES TO REVIEWER #3

---***---***---***---***---***---***---***---***---

Reviewer #3: The study was carried out robustly and presented in an intelligible manner. It would benefit from providing more information about the participants in terms of whether or not they had previous experience abroad or with the countries mentioned. 

Thank you very much for your comment and suggestion. All participants visited the respective destination countries for the first time at the occasion of the STSA. Though they were not able to communicate using any of the languages spoken locally, they could at a basic level speak and write in English. Self-reports indicated that STSA students had none or very little experience overseas. This information was added to the Methods section of the revised manuscript.

We hope all your questions and concerns were properly addressed in the revised manuscript. Thank you very much for your time and effort in improving our paper. We look forward to hearing from you.

With very best wishes,

T. Yamanaka, on behalf of the authors.

---

## [Decision Letter · Decision Letter 1]

26 Oct 2021

PONE-D-21-12484R1Assessing changes in mood state in university students following short-term study abroadPLOS ONE

Dear Dr. Yamanaka

Thank you for submitting your manuscript to PLOS ONE. After careful consideration, we feel that it has merit but does not fully meet PLOS ONE’s publication criteria as it currently stands. Therefore, we invite you to submit a revised version of the manuscript that addresses the points raised during the review process.

You should consider the new suggestions from Reviewer 1 and also incorporate the descriptive statistics requested by Reviewer 2.

We look forward to receiving your revised manuscript.

Kind regards,

Berta Schnettler

Academic Editor

PLOS ONE

Journal Requirements:

Reviewers' comments:

Reviewer's Responses to Questions

**Comments to the Author**

1. If the authors have adequately addressed your comments raised in a previous round of review and you feel that this manuscript is now acceptable for publication, you may indicate that here to bypass the “Comments to the Author” section, enter your conflict of interest statement in the “Confidential to Editor” section, and submit your "Accept" recommendation.

Reviewer #1: (No Response)

Reviewer #2: All comments have been addressed

Reviewer #3: All comments have been addressed

2. Is the manuscript technically sound, and do the data support the conclusions?

Reviewer #1: Yes

Reviewer #2: Yes

Reviewer #3: (No Response)

3. Has the statistical analysis been performed appropriately and rigorously? 

Reviewer #1: Yes

Reviewer #2: Yes

Reviewer #3: (No Response)

4. Have the authors made all data underlying the findings in their manuscript fully available?

Reviewer #1: Yes

Reviewer #2: Yes

Reviewer #3: (No Response)

5. Is the manuscript presented in an intelligible fashion and written in standard English?

Reviewer #1: Yes

Reviewer #2: Yes

Reviewer #3: (No Response)

6. Review Comments to the Author

Reviewer #1: Although I appreciate the authors’ careful attention to each aspect of the reviewers’ comments, I remain concerned about the rationale for studying mood in STSA. In their revision, the authors have added a discussion of Lysgaard’s U-curve hypotheses (in the Introduction and the Discussion sections) as central to their rationale. Yet, empirical research has not supported the U-curve hypothesis (see, for example, Chien, 2016; Gray & Savicki, 2015). As I read the revision, I found myself still wondering about the possible significance and implications of the authors’ findings on STSA participants’ elevated mood. The statement in the Discussion indicating that “This is important because short-term programs are a more accessible alternative to long term…” still didn’t answer my question. Then, in the last paragraph of the manuscript, the authors make a fascinating comment – “[these findings] suggest that the SA participants did not suffer a reverse culture shock in the aftermath of the STSA, in contrast with what has been reported about long-term SA program participants.” It seems to me that this latter point is the more significant contribution of this study. If, in fact, STSA programs are characterized by less severe re-entry shock than their longer counterparts, this information could contribute to our understanding of why re-entry shock occurs and may shape the content of pre-departure and in-country orientation for STSA participants. To my knowledge, re-entry shock in STSA is a topic that has received little attention in terms of empirical investigations. I suggest one more revision in which any reference to the U-curve is removed and emphasis is placed on the issue of re-entry shock (as indicated by the assessment of mood). I believe that doing so could significantly strengthen this contribution to the study abroad literature.

Chien, Y.-Y. G. (2016). After six decades: Applying the U-curve hypothesis to the adjustment of international postgraduate students. Journal of Research in International Education, 15(1), 32-51. https://doi.org/10.1177%2F1475240916639398

Gray, K. M., & Savicki, V. (2015). Study abroad reentry: Behavior, affect, and cultural distance. Frontiers: The Interdisciplinary Journal of Study Abroad, 26, 264-278. https://doi.org/10.36366/frontiers.v26i1.370

Reviewer #2: The authors have addressed most of my concerns and the revised paper is much clearer. Although the author refused to provide descriptive statistics, I am fine with that as long as the editor and the journal don't require that. The paper will contribute to the field of SA and I recommend it for publication.

Reviewer #3: (No Response)

7. PLOS authors have the option to publish the peer review history of their article (what does this mean?). If published, this will include your full peer review and any attached files.

Reviewer #1: No

Reviewer #2: No

Reviewer #3: No

---

## [Author Response · Author response to Decision Letter 1]

25 Nov 2021

---***---***---***---***---***---***---***---***---

COMMENTS FROM/REPLIES TO REVIEWER #1

---***---***---***---***---***---***---***---***---

Reviewer #1: 

Although I appreciate the authors’ careful attention to each aspect of the reviewers’ comments, I remain concerned about the rationale for studying mood in STSA. In their revision, the authors have added a discussion of Lysgaard’s U-curve hypotheses (in the Introduction and the Discussion sections) as central to their rationale. Yet, empirical research has not supported the U-curve hypothesis (see, for example, Chien, 2016; Gray & Savicki, 2015). As I read the revision, I found myself still wondering about the possible significance and implications of the authors’ findings on STSA participants’ elevated mood. The statement in the Discussion indicating that “This is important because short-term programs are a more accessible alternative to long term…” still didn’t answer my question. Then, in the last paragraph of the manuscript, the authors make a fascinating comment – “[these findings] suggest that the SA participants did not suffer a reverse culture shock in the aftermath of the STSA, in contrast with what has been reported about long-term SA program participants.” It seems to me that this latter point is the more significant contribution of this study. If, in fact, STSA programs are characterized by less severe re-entry shock than their longer counterparts, this information could contribute to our understanding of why re-entry shock occurs and may shape the content of pre-departure and in-country orientation for STSA participants. To my knowledge, re-entry shock in STSA is a topic that has received little attention in terms of empirical investigations. I suggest one more revision in which any reference to the U-curve is removed and emphasis is placed on the issue of re-entry shock (as indicated by the assessment of mood). I believe that doing so could significantly strengthen this contribution to the study abroad literature.

Chien, Y.-Y. G. (2016). After six decades: Applying the U-curve hypothesis to the adjustment of international postgraduate students. Journal of Research in International Education, 15(1), 32-51. https://doi.org/10.1177%2F1475240916639398

Gray, K. M., & Savicki, V. (2015). Study abroad reentry: Behavior, affect, and cultural distance. Frontiers: The Interdisciplinary Journal of Study Abroad, 26, 264-278. https://doi.org/10.36366/frontiers.v26i1.370

We thank the Reviewer for the useful suggestions made for improving our manuscript. As requested, we have now added arguments and supporting material about reverse culture shock after STSA, both in the Introduction and Discussion. Additionally, we now cite the empirical evidence that does not support Lysgaard’s U-curve hypotheses in the Introduction, and reference to the U-curve has been removed from the Discussion. In addition to citing Chien (2016) and Gray et al (2015), three other references were added to the manuscript. References added are as follows:

[22] Gullahorn JT, Gullahorn JE. An Extension of the U-Curve Hypothesis. Journal of Social Issues. 1963;19(3):33-47.

→ Cited in the 7th paragraph of "Introduction".

[23] Ward C, Okura Y, Kennedy A, Kojima T. The U-Curve on trial: A longitudinal study of psychological and sociocultural adjustment during the Cross-Cultural transition. Cross-cultural transition. 1998.

→ Cited in the 7th paragraph of "Introduction".

[24] Chien Y-YG. After six decades: Applying the U-curve hypothesis to the adjustment of international postgraduate students. Journal of Research in International Education. 2016;15(1):32-51.

→ Cited in the 7th paragraph of "Introduction".

[25] Martin JN, Harrell T. Intercultural reentry of students and professionals: theory and practice. In Handbook of Intercultural Training. Publications Inc. 2004. p. 309-336

→ Cited in the 7th paragraph of "Introduction".

[26] Gray KM, Savicki V. Study Abroad Reentry: Behavior, Affect, and Cultural Distance. Frontiers: The Interdisciplinary Journal of Study Abroad. 2015 ;26(1).

→ Cited in the 7th paragraph of "Introduction".

→ Cited in the 9th paragraph of "Discussion".

We hope all your suggestions and concerns were properly addressed in the revised manuscript. Thank you very much for your time and effort in improving our paper. 

---***---***---***---***---***---***---***---***---

COMMENTS FROM/REPLIES TO REVIEWER #2

---***---***---***---***---***---***---***---***---

Reviewer #2: The authors have addressed most of my concerns and the revised paper is much clearer. Although the author refused to provide descriptive statistics, I am fine with that as long as the editor and the journal don't require that. The paper will contribute to the field of SA and I recommend it for publication.

Thank you for recommending our manuscript for publication in PLOS ONE. Note that all raw data is provided, and in the main text we have reported rigorous and appropriate statistical analyses of all results, which we believe are in line with Journal policy. 

--***---***---***---***---***---***---***---***---

MINOR CORRECTIONS

--***---***---***---***---***---***---***---***---

A minor correction was made to the name of ethics committee. Plus we now make it explicit that ‘written’ informed consent was obtained prior to participation in the study.

---

## [Decision Letter · Decision Letter 2]

10 Dec 2021

Assessing changes in mood state in university students following short-term study abroad

PONE-D-21-12484R2

Dear Dr. Yamanaka,

We’re pleased to inform you that your manuscript has been judged scientifically suitable for publication and will be formally accepted for publication once it meets all outstanding technical requirements.

Kind regards,

Berta Schnettler

Academic Editor

PLOS ONE

Additional Editor Comments (optional):

Reviewers' comments:

Reviewer's Responses to Questions

**Comments to the Author**

1. If the authors have adequately addressed your comments raised in a previous round of review and you feel that this manuscript is now acceptable for publication, you may indicate that here to bypass the “Comments to the Author” section, enter your conflict of interest statement in the “Confidential to Editor” section, and submit your "Accept" recommendation.

Reviewer #1: All comments have been addressed

Reviewer #2: All comments have been addressed

2. Is the manuscript technically sound, and do the data support the conclusions?

Reviewer #1: Yes

Reviewer #2: Yes

3. Has the statistical analysis been performed appropriately and rigorously? 

Reviewer #1: (No Response)

Reviewer #2: Yes

4. Have the authors made all data underlying the findings in their manuscript fully available?

Reviewer #1: (No Response)

Reviewer #2: Yes

5. Is the manuscript presented in an intelligible fashion and written in standard English?

Reviewer #1: (No Response)

Reviewer #2: Yes

6. Review Comments to the Author

Reviewer #1: (No Response)

Reviewer #2: The authors addressed all my comments and the revised version now reads much clearer. Therefore, I am happy to recommend it for publication.

7. PLOS authors have the option to publish the peer review history of their article (what does this mean?). If published, this will include your full peer review and any attached files.

Reviewer #1: No

Reviewer #2: No

---

## [Editor Report · Acceptance letter]

14 Dec 2021

PONE-D-21-12484R2 

Assessing changes in mood state in university students following short-term study abroad 

Dear Dr. Yamanaka:

I'm pleased to inform you that your manuscript has been deemed suitable for publication in PLOS ONE. Congratulations! Your manuscript is now with our production department. 

Kind regards, 

on behalf of

Dr. Berta Schnettler 

Academic Editor

PLOS ONE